# Acupuncture for Behavioral and Psychological Symptoms of Dementia: A Systematic Review and Meta-Analysis

**DOI:** 10.3390/jcm10143087

**Published:** 2021-07-13

**Authors:** Chan-Young Kwon, Boram Lee

**Affiliations:** 1Department of Oriental Neuropsychiatry, Dong-eui University College of Korean Medicine, 52-57 Yangjeong-ro, Busanjin-gu, Busan 47227, Korea; 2Department of Clinical Korean Medicine, Graduate School, Kyung Hee University, 26 Kyunghee-daero, Dongdaemun-gu, Seoul 02447, Korea; qhfka9357@naver.com

**Keywords:** dementia, BPSD, acupuncture therapy, acupuncture, systematic review

## Abstract

Dementia is an important health issue worldwide, and non-pharmacological strategies for the management of behavioral and psychological symptoms of dementia (BPSD) are considered to be important. This review analyzes the effectiveness and safety of acupuncture for BPSD. Thirteen electronic databases were comprehensively searched to find clinical studies using acupuncture on BPSD, published up to December 2020. Five randomized controlled clinical trials and two before-after studies, mainly on Alzheimer’s disease (AD), were included. Meta-analysis suggested that the total effective rate based on BPSD symptoms in the acupuncture combined with psychotropic drugs group was significantly higher than that in the psychotropic drugs group (risk ratio, 1.27; 95% confidence interval, 1.11 to 1.45; I^2^ = 51%). In terms of other outcomes related to BPSD, acupuncture as an adjunctive therapy, but not as monotherapy, was associated with significant benefits in most included studies. However, the included studies did not have optimal methodological quality. Our review highlights the limited evidence proving the effectiveness and safety of acupuncture for BPSD in patients with AD. Although some clinical studies have reported the potential benefits of adjuvant acupuncture in managing BPSD, the evidence is not robust and is based on small studies. Therefore, high-quality research in this field is needed.

## 1. Introduction

Dementia refers to a clinical syndrome that significantly impairs an individual’s activities of daily living (ADL) with gradual cognitive decline, and it is an important health issue worldwide [1]. Along with strategies to prevent the occurrence of dementia based on an understanding of its multifactor pathology, strategies to reduce the burden of dementia patients and their caregivers as well as the socioeconomic burden is becoming increasingly important [1]. According to estimates, the number of patients with Alzheimer’s disease (AD), a representative type of dementia, is expected to be 13.8 million in the United States and 115.4 million globally as of 2050 [2,3].

Behavioral and psychological symptoms of dementia (BPSD), which most patients with dementia experience, are thought to be the main cause of the dementia-related burden [4,5]. Therefore, the management of BPSD is a promising strategy to reduce the burden of dementia, and a non-pharmacological approach is particularly preferred in terms of benefit-risk ratio [6,7]. However, psychosocial interventions for BPSD are still not implemented systematically, and pharmacological interventions that can be associated with a number of negative health outcomes are being used [8,9,10]. As many non-pharmacological interventions are difficult to implement in actual clinical settings due to barriers related to resident unavailability and external barriers [11], it is necessary to explore non-pharmacological interventions that are individualized, can be easily implemented, are accessible, and are effective and safe for patients with dementia, in terms of evidence-based medicine (EBM).

Acupuncture, a promising non-pharmacological intervention, is recommended worldwide for some conditions such as pain [12] although it is not evaluated to have robust evidence to be officially recommended for the management of BPSD [13]. Acupuncture has been studied not only for pain conditions, but also for a wide range of health conditions such as neoplasms/cancer, pregnancy or labor, mood disorders, stroke, nausea/vomiting, sleep, and paralysis/palsy [14], and has the potential to improve overall health [15]. In addition, this modality appears to be generally safe [16]. In this context, acupuncture can be considered a promising treatment for the elderly population. Psychiatric manifestations in patients with dementia have been well documented in neuroimaging and neuropathology studies [17]. The effects of acupuncture on the nervous system of human subjects and animal models with neurological diseases including AD have also been reported [18]. It has been stated that acupuncture can influence AD pathology through the inhibition of accumulation of toxic proteins, regulation of glucose metabolism, reduction of neuronal apoptosis, and neuroprotective effects [18]. Therefore, this treatment may have a therapeutic effect on BPSD, which accompanies AD pathology [19,20].

However, there have been insufficient studies to examine the applicability of acupuncture for BPSD by comprehensively and systematically collecting previously published literature. Examining the applicability of acupuncture from the perspective of EBM potentially improves the management of dementia, thereby contributing to the reduction of the burden caused by dementia. Therefore, this systematic review aimed to analyze the effectiveness, safety, and research status of acupuncture in BPSD management.

## 2. Materials and Methods

The protocol of this systematic review was registered in the OSF registries (URL: https://osf.io/hu5ac) (Accessed on 30 November 2020) and the International Prospective Register of Systematic Reviews (registration number: CRD42020211005) before the start of the study. The protocol for this study was published in an international academic journal [21]. This systematic review complied with the Preferred Reporting Items for Systematic Reviews and Meta-Analyses (PRISMA) statement (Appendix A) [22].

### 2.1. Information Sources and Search Strategy

A comprehensive study search was conducted in 13 electronic databases, including Medline via PubMed, EMBASE via Elsevier, the Cochrane Central Register of Controlled Trials, Allied and Complementary Medicine Database via EBSCO, Cumulative Index to Nursing and Allied Health Literature via EBSCO, PsycARTICLES via ProQuest, Oriental Medicine Advanced Searching Integrated System, Koreanstudies Information Service System, Research Information Service System, Korean Medical Database, Korea Citation Index, China National Knowledge Infrastructure, and Wanfang Data on December 28, 2020. All studies published from their inception to the search date were considered. To find additional gray literature such as theses or conference abstracts, relevant literature reference lists and trial registries, including clinicaltrials.gov, were reviewed. In addition, consultations with experts in this area were conducted. The study search process was conducted by a single researcher (BL). The search strategies and results in each database are described in Appendix A.

### 2.2. Eligibility Criteria

The inclusion criteria for this review were as follows: (1) *Population*: Patients with dementia regardless of the type in long-term care facilities, community, or specialized geriatric assessment and psychiatric units was allowed. Only patients with dementia, with standard diagnostic criteria or validated tools, were included. The standard diagnostic criteria for dementia included the Diagnostic and Statistical Manual of Mental Disorders, the International Classification of Diseases, the National Institute of Neurological and Communicative Disorders and Stroke and the Alzheimer’s Disease and Related Disorders Association, and other recommended diagnostic criteria. There were no restrictions on sex, age, or race/ethnicity of the patients. (2) *Intervention*: Acupuncture regardless of the type as monotherapy or adjunctive therapy to psychotropic drugs such as anxiolytics, antidepressants, and antipsychotics, with or without routine care for dementia as treatment interventions were included. Studies that did not list details such as the treatment period and treatment points (i.e., acupoints) were excluded. In addition, studies involving psychotherapy as treatment or control interventions were excluded. (3) *Comparator*: Wait-list, placebo (sham-acupuncture), or psychotropic drugs such as anxiolytics, antidepressants, and antipsychotics, with or without routine care for dementia, as control interventions were allowed. (4) *Outcome*: The severity of BPSD symptoms, such as Behavior Pathology in Alzheimer Disease Rating Scale (BEHAVE-AD) [23], Neuropsychiatric Inventory (NPI) [24], and Brief Psychiatric Rating Scale (BPRS) [25] were considered as the primary outcomes. The secondary outcome measures included the total effective rate (TER) for BPSD symptoms; ADL of patients such as Barthel Index [26]; instrumental ADL such as ADL Prevention Instrument [27]; quality of life (QOL) of patients with dementia such as Alzheimer Disease Related Quality of QOL [28]; caregiver burden of caregivers such as Caregiver Burden Inventory [29]; QOL of caregivers such as the Short Form 36 Health Survey (SF-36) [30]; placement in long-term care facility from home; and safety data such as incidence of adverse events (AEs). (5) *Study design*: clinical studies, regardless of the type, such as randomized controlled clinical trials (RCTs), non-randomized controlled clinical trials, and before-after studies. There were no restrictions on the publication language or publication status.

### 2.3. Study Selection

All documents retrieved from the study search process were imported into EndNote X8 (Clarivate Analytics, Philadelphia, PA, USA). After removing duplicates, the titles and abstracts of the documents were screened for inclusion. After the initial screening, the full texts of the remaining documents were carefully reviewed to evaluate the final inclusion. The study search process was conducted by two independent researchers (C.Y.K. and B.L.). Any disagreements between the two researchers were resolved through a consensus.

### 2.4. Data Extraction

In the data extraction process, a pre-defined form in Excel 2016 (Microsoft, Redmond, WA, USA) was used by two independent researchers (C.Y.K. and B.L.). The following items were extracted from the included studies: first author’s name, publication year, country of publication, sample size and withdrawals, details of participants, treatment and control intervention, duration of intervention, outcome measures and results, AEs, and information for assessing the risk of bias (RoB). In particular, details of acupuncture procedures, such as acupoints, stimulation method, and needle retention time, were extracted. The authors contacted the corresponding authors of the original studies via e-mail, when the data in each included study were insufficient or ambiguous. Any disagreements between the two researchers were resolved through a consensus.

### 2.5. Risk of Bias Assessment

The RoB evaluation tools of the included studies were applied differently according to the study type. The inclusion criteria of the systematic review allowed all original clinical studies regardless of the study type, but the types included were limited to RCTs and before-after studies. In the case of RCTs, the Cochrane Collaboration’s Rob tool was used to evaluate RoB. Using this tool, the RoB of RCTs can be evaluated by categorizing it into random sequence generation, allocation concealment, blinding of participants, personnel, and outcome assessors, completeness of outcome data, selective reporting, and other biases. For other biases, a statistical baseline imbalance of participants’ mean age, sex, disease period, or disease severity between the treatment and control groups was considered. Each domain was evaluated as “low risk,” “unclear risk,” or “high risk,” and the evaluation method of the Cochrane Handbook was followed [31]. The results of the RoB assessment were presented as figures using Review Manager software (version 5.4; Cochrane, London, UK). In case of before-after studies, the Quality Assessment Tool for Before-After (Pre-Post) Studies With No Control Group produced by the US National Heart Lung and Blood Institute was used [32]. The RoB assessment process was conducted by two independent researchers (C.Y.K. and B.L.), and any disagreements between the two researchers were resolved through consensus.

### 2.6. Data Synthesis and Analysis

A qualitative analysis of all included studies, including demographic characteristics of participants, details of interventions, and outcomes, was conducted. Quantitative synthesis was performed with the outcome measures if there were two or more studies using the same type of treatment and control interventions. The meta-analysis was conducted and presented as forest plots using Review Manager software (version 5.4; Cochrane, London, UK). For continuous outcomes, the mean differences with 95% confidence intervals (CIs) were calculated. For binary outcomes, the risk ratios (RRs) with 95% CIs were calculated. Heterogeneity between the studies in terms of effect measures was assessed using both the χ^2^ test and the I^2^ statistic, and I^2^ values greater than 50% and 75% were interpreted as substantial and considerable heterogeneity, respectively. In the meta-analysis, a random-effects model was used if included studies had significant heterogeneity (an I^2^ > 50%), while a fixed-effect model was used if the heterogeneity was not significant or if the number of studies included in the meta-analysis was less than 5 [33,34]. According to the protocol [21], subgroup analyses based on severity of dementia, type of dementia, severity of BPSD, and treatment duration were planned; however, these were not conducted because the number of studies included and the relevant data were insufficient. Sensitivity analysis in meta-analyses was planned to remove studies with high RoB and outliers that are numerically distant from the rest of the data, but this was not conducted for the same reason.

### 2.7. Publication Bias

Assessment of publication bias using a funnel plot was planned in the protocol [21]; however, the assessment was not conducted, as there were no more than 10 studies included in each meta-analysis.

## 3. Results

### 3.1. Study Selection

A total of 8715 documents were identified through initial database searches, and 1600 duplicates were excluded. After reviewing the titles and abstracts of the remaining 7115 documents, 44 potentially relevant articles were selected. The full texts of the 44 articles were assessed for the final inclusion. Among them, the following numbers of studies were excluded: not clinical studies (*n* = 3), not report the standard diagnostic criteria or validated tools for participants (*n* = 3), not on acupuncture (*n* = 1), comparing two different kinds of acupuncture (*n* = 2), not reporting outcomes related to BPSD (*n* = 18), without details about the study (*n* = 5), using same data (thesis or conference abstract) (*n* = 3), using the same data (published in another journal) (*n* = 1), and not available full-text (*n* = 1) (Appendix A). Finally, a total of seven studies (five RCTs [35,36,37,38,39] and two before-after studies [40,41]) were included in this review (Figure 1).

### 3.2. Study Characteristics

Among the seven included studies, six [35,36,37,38,39,41] were conducted in China, and the remaining study [40] was conducted in America. As for the type of publication, there was one thesis [41], and the remainder [35,36,37,38,39,40] were journal articles. As for the type of dementia, except for one study [40] involving patients with AD or vascular dementia (VD), all studies [35,36,37,38,39,41] were in patients with AD. As for the type of acupuncture, three [37,40,41] used manual acupuncture, three [35,38,39] used electroacupuncture, and one [36] used scalp acupuncture.

Among the five RCTs [35,36,37,38,39], there were three [35,36,38] comparing acupuncture combined with psychotropic drugs and psychotropic drugs alone, one [37] comparing acupuncture and anti-dementia drugs, and one [39] comparing acupuncture combined with routine care and routine care alone. No study recruited participants according to pattern identification. The treatment period ranged from 4 weeks to 12 years, of which 8 weeks was the most common in the two RCTs [35,36]. Only one RCT [37] conducted a follow-up assessment after completion of treatment. Two RCTs [37,39] were approved by the institutional review board for the study, and three RCTs [37,38,39] received consent forms from the participants. The two before-after studies used manual acupuncture for 9–12 weeks [40] and 8 weeks [41], respectively. In addition, the two studies [40,41] also did not use pattern identification in recruiting participants. The two studies [40,41] did not describe approval by the institutional review board for the study; however, they reported that they received consent forms from participants (Table 1 and Table 2).

### 3.3. Risk of Bias in Studies

For RCTs, a total of four studies [36,37,38,39] with proper randomization methods, such as using a random number table, were evaluated as low risk in the random sequence generation domain. Only one study [37] used a special envelope to perform allocation concealment, which was assessed to have low risk in the domain of allocation concealment. None of the included studies used sham acupuncture as a control, suggesting that double blinding was impossible. Therefore, all studies were evaluated as having high risk in the domain of performance bias. Only one study [37] described the blinding of the outcome assessor, and the domain of detection bias was evaluated as having a low risk of bias. All studies were evaluated as low risk in the domain of incomplete outcome data, because there were no dropouts [35,36,38,39] or the number of dropouts was not expected to have a significant effect on the results because it was small [37]. Most studies [36,37,38,39] were evaluated as having a low risk of reporting bias in the domain of selective reporting using an objective and validated evaluation tool related to BPSD. However, one study [35] did not present the results of some outcomes as numerical values, so the domain was evaluated as high risk. As all studies reported baseline statistical homogeneity of participants’ mean age, sex, disease period, or disease severity, the domains of other sources of bias were evaluated as low risk (Figure 2). In two before-after studies [40,41], both clearly stated the study questions, eligibility criteria for the study population, and outcome measures. In addition, these studies [40,41] were lost to follow-up after a baseline of 20% or less, and a statistical test using *p*-value was performed, and two or more results of assessment after baseline were reported. However, both studies [40,41] were small with a sample size of less than 60 [42], there was no previously published protocol, no assessor blinding was reported, and individual-level data were not considered. In addition, one study [40] lacked detailed descriptions related to the acupuncture method, such as stimulation method and needle retention time (Appendix A).

### 3.4. Effectiveness and Safety of Acupuncture in Included RCTs

#### 3.4.1. Acupuncture as a Monotherapy

Jia [37], comparing manual acupuncture and donepezil (first 4 weeks: 5 mg/day; thereafter: 10 mg/day) in 87 AD patients for 12 weeks, reported that there was no statistically significant difference using the AD Cooperative Study—ADL, 23-item scale (after 12 weeks: 47.85 (mean) ± 11.22 (standard deviation) vs. 48.20 ± 13.16, *p* > 0.05; 12-week follow-up: 48.18 ± 11.32 vs. 49.43 ± 13.45, *p* > 0.05) and NPI total score (after 12 weeks: 7.25 ± 2.69 vs. 7.61 ± 2.30, *p* > 0.05; 12-week follow-up: 8.13 ± 2.78 vs. 9.31 ± 2.42, *p* > 0.05). There were five cases of AE in the acupuncture group, including four cases of punctate hemorrhage and one case of bruising. In the donepezil group, there were seven cases of AE, including dizziness, nausea, loss of appetite, diarrhea, constipation, fatigue, and agitation. Interestingly, the authors reported that the acupuncture group unexpectedly improved the following symptoms: insomnia (five cases), constipation (four cases), elderly male patients with benign prostatic hyperplasia (six cases), and knee arthritis (two cases).

#### 3.4.2. Acupuncture as an Adjunctive Therapy

Meta-analysis was possible only for TER for BPSD symptoms. As a result, TER in the acupuncture combined with psychotropic drugs group was significantly higher than that in the psychotropic drugs group (three studies [35,36,38]; RR, 1.27; 95% CI, 1.11 to 1.45; *I*^2^ = 51%) (Figure 3).

Ou [35], comparing electroacupuncture combined with perphenazine (4-30 mg/day) and perphenazine (8–40 mg/day) in 30 AD patients with BPSD for 8 weeks, reported no statistically significant difference on BPRS between the groups (27.14 ± 7.91 vs. 28.23 ± 8.42, *p* > 0.05). There was no statistically significant difference in the Rating Scale for Extrapyramidal Side Effects score after 1 week of treatment (1.3 ± 1.0, 2.1 ± 0.4, *p* > 0.05). Huang (2014) [36], comparing scalp acupuncture combined with fluoxetine (20 mg/day) and fluoxetine (20 mg/day) alone in 100 AD patients with depression for 8 weeks, reported that compared to the control group, the treatment group showed a statistically significantly lower Hamilton depression rating scale (HAMD) score (9.32 ± 4.93 vs. 11.89 ± 5.97, *p* < 0.05) and higher ADL score (36.76 ± 3.29 vs. 34.92 ± 4.33, *p* < 0.05). Zhang [38] compared electroacupuncture combined with midazolam (7.5 mg/day) and midazolam (7.5 mg/day) alone in 82 AD patients with sleep disorder for 30 days and reported that compared to the control group, the treatment group showed a statistically significantly lower global score of the Pittsburgh sleep quality index (PSQI) (1.59 ± 2.15 vs. 4.15 ± 1.77, *p* < 0.05).

Zhao [39], comparing electroacupuncture combined with passive music therapy and routine care (i.e., routine nursing, psychological comfort, daily cognitive training, diet modification) and routine care in 120 AD patients for 4 weeks, reported that compared to the control group, the treatment group showed a statistically significantly higher ADL score (64.92 ± 8.29 vs. 47.07 ± 10.18, *p* < 0.05) and BEHAVE-AD total score (37.45 ± 4.40 vs. 27.21 ± 4.20, *p* < 0.05). Moreover, they found that all subscales of SF-36 for the treatment group were statistically significantly higher than that of control group, including physical functioning (71.39 ± 6.58 vs. 55.13 ± 7.14, *p* < 0.05), body pain (73.22 ± 7.82 vs. 59.61 ± 7.31, *p* < 0.05), general health perception (68.77 ± 6.16 vs. 50.33 ± 5.24, *p* < 0.05), vitality (71.82 ± 6.64 vs. 55.64 ± 5.67, *p* < 0.05), social functioning (80.27 ± 7.89 vs. 69.55 ± 8.10, *p* < 0.05), emotional role functioning (71.46 ± 6.21 vs. 60.24 ± 6.59, *p* < 0.05), and mental health (73.62 ± 7.46 vs. 62.31 ± 5.88, *p* < 0.05).

### 3.5. Results in Included Before-After Studies

Lombardo [40], performing manual acupuncture in 11 patients with AD or VD for 9-12 weeks, reported that the anxiety subscale of the Profile of Mood States (POMS) (8.8 ± 6.3 to 4.6 ± 3.4, *p* = 0.05), state anxiety of State-Trait Anxiety Inventory (STAI) (49.5 ± 8.4 to 40.1 ± 8.0, *p* = 0.005), and Cornell Scale for Depression in Dementia (CSDD) (6.4 ± 5.0 to 3.1 ± 3.0, *p* = 0.011) were significantly improved. In addition, some subscales of SF-36 in caregivers were significantly improved, including vitality (4.1 ± 1.2 to 3.4 ± 0.9, *p* = 0.003) and anxiety (3.2 ± 0.7 to 2.5 ± 0.8, *p* = 0.006), but not depression (2.8 ± 0.7 to 2.7 ± 1.0, *p* = 0.394). There was no statistically significant difference in the Geriatric Depression Scale (GDS) (7.4 ± 3.9 to 6.7 ± 7.0, *p* = 0.358) and subscales of SF-36 in participants, including vitality (2.7 ± 0.8, 2.6 ± 0.9, *p* = 0.416), anxiety (2.5 ± 0.7 to 2.1 ± 0.6, *p* = 0.441), and depression (2.6 ± 1.0 ± 2.2 ± 0.7, *p* = 0.350).

Ying [41], performing manual acupuncture in 33 AD patients with BPSD for 8 weeks, reported that the BEHAVE-AD total score (18.90 ± 6.67 to 15.37 ± 7.42, *p* < 0.001), and some subscales including paranoid and delusional ideation (5.22 ± 2.60 to 4.51 ± 2.50, *p* < 0.01), activity disturbances (3.80 ± 2.11 to 3.38 ± 2.28, *p* = 0.002), affective disturbances (2.17 ± 1.34 to 1.27 ± 0.98, *p* < 0.01), and anxieties and phobias (3.00 ± 1.88 to 1.50 ± 1.43, *p* < 0.01) were statistically significantly improved. However, other subscales of BEHAVE-AD including hallucinations (1.40 ± 1.43 to 1.43 ± 1.30, *p* = 0.12), aggressiveness (1.43 ± 1.83 to 1.50 ± 1.85, *p* = 0.53), and diurnal rhythm disturbances (1.90 ± 0.93 to 2.00 ± 0.98, *p* = 0.58) as well as ADL (47.67 ± 8.50 to 47.03 ± 9.91, *p* = 0.141) showed no statistically significant difference.

## 4. Discussion

### 4.1. Summary of Evidence

This systematic review is the most comprehensive review and meta-analysis to date conducted to analyze the effectiveness, safety, and research status of acupuncture for BPSD. A total of seven clinical studies, including five RCTs [35,36,37,38,39] and two before-after studies [40,41], mainly on patients with AD, were included in this review. In these studies, RCTs were analyzed to evaluate the effectiveness and safety of BPSD. On the other hand, the before-after studies were used as the basis for understanding the current status of research in this field, considering the low level of evidence. Meta-analysis was only possible for TER based on BPSD symptoms from three RCTs [35,36,38]. According to the meta-analysis results, acupuncture combined with psychotropic drugs showed statistically significant superiority in terms of TER compared to the psychotropic drugs alone group. For the other outcomes, only qualitative synthesis was conducted. First, in terms of outcomes related to BPSD, acupuncture as an adjunctive therapy, but not as monotherapy, was associated with significant benefits in most included studies. One study [35] compared electroacupuncture combined with perphenazine and perphenazine and reported that there was no significant difference in the effect of the two interventions on the BPRS score. However, considering that the dose of perphenazine was used less in the combined therapy group (4–30 mg/day) than in the control group (8–40 mg/day), the results of this study suggest that EA may be helpful in BPSD management. Similarly, the other two RCTs, comparing acupuncture combined with psychotropic drugs and psychotropic drugs, found the significant benefits of acupuncture as an adjunctive therapy for HAMD [36] and PSQI [38]. One RCT [39], which investigated the additional benefits of acupuncture and music therapy based on routine care, reported the BEHAVE-AD score as a BPSD-related outcome, but the value was suspicious. The BEHAVE-AD score increased significantly after treatment in both groups, but the authors explained that BPSD symptoms were reduced in this study. Given that higher BEHAVE-AD scores reflect more severe symptoms of BPSD [23], these explanations seem unusual; therefore, we contacted the corresponding author of the paper via e-mail, but did not receive a reply. Second, in terms of ADL, two RCTs reported additional benefits of acupuncture as an adjunctive therapy to midazolam [36] or routine care [39]. However, acupuncture as a monotherapy did not show statistically significant superiority in ADL compared to donepezil [38]. Third, in one RCT [39], the patient’s QOL was evaluated using the SF-36. The results showed that acupuncture and music therapy added to routine care significantly improved participants’ QOL in terms of physical functioning, body pain, general health perception, vitality, social functioning, emotional functioning, and mental health. Fourth, only two RCTs [35,37] reported the safety data of the interventions. Acupuncture was mainly associated with mild local AEs such as punctate hemorrhage and bruising and did not have a significant effect on safety concerns of drug treatment such as extrapyramidal side effects. On the other hand, there were two before-after studies [40,41] included in this review, both of which used manual acupuncture, and significant improvements were reported in BPSD-related outcomes such as POMS, STAI, CSDD, and BEHAVE-AD. However, no significant improvement was observed in the GDS, the anxiety and depression subscales of SF-36, and hallucinations, aggressiveness, and diurnal rhythm disturbance subscales of BEHAVE-AD. In addition, one study [41] that reported ADL found no statistically significant change. In summary, in the included studies, improvements in individual BPSD after acupuncture were reported, in particular, for depression, anxiety, sleep quality, paranoid and delusional ideation, and activity disturbances. Among them, the improved individual symptoms of BPSD after acupuncture in the included RCTs were depression and sleep quality.

Overall, the included studies did not have optimal methodological quality. In particular, most studies were small-scale, and few studies reported blinding of outcome assessors. Considering that blinding of participants and personnel in the perfect sense is almost impossible due to the nature of acupuncture, the lack of blinding in outcome assessors could potentially affect the reliability of each study result. These blinding issues and small-scale limitations also apply to the two before-after studies.

### 4.2. Implications of the Results

This review presents the most comprehensive literature review conducted to evaluate the effectiveness and safety of acupuncture for BPSD. Although studies have shown that acupuncture, primarily as a form of adjuvant therapy in patients with AD, has the potential to improve overall BPSD as well as individual BPSD symptoms such as depression and sleep quality, these results were not supported by a sufficient number of studies or evidence of sufficient quality. However, given that the establishment of a non-pharmacological approach to the management of BPSD is important [6,7] and that effective and safe non-pharmacological strategies should be implemented in a viable manner [8,9,10], acupuncture may still be an attractive intervention. This is because the potential of acupuncture has been continuously reported for conditions, especially in the elderly, regarding the limitations of conventional pharmacological treatment, such as poly-pharmacy [43,44].

Although the underlying therapeutic mechanism of acupuncture for BPSD in patients with AD has not been fully elucidated, the results of pre-clinical and clinical experiments can be referred to. Lu that manual acupuncture on ST36 for 30 days in a rat AD model activated brain areas including the orbital cortex, medulla oblongata, and pontine tegmentum [45]. Moreover, Shan found that manual acupuncture on LI4 and LR3 in patients with AD was related to enhanced activations in cognitive-related areas, including the inferior frontal gyrus, as well as sensorimotor-related areas, the basal ganglia, and the cerebellum, by using functional magnetic resonance imaging (fMRI) [46]. A review investigating changes in acupuncture-related brain activity in fMRI concluded that acupuncture stimuli could promote activity changes in a wide range of brain regions, including the somatosensory cortices, limbic system, basal ganglia, brain stem, and cerebellum [47]. Acupuncture-induced changes in local brain activity in AD patients may be related to the therapeutic effect of acupuncture on BPSD. Among them, the possible impact of acupuncture on the orbitofrontal cortex seems to be promising. A previous systematic review suggested that volume reductions or decreased metabolism in the orbital cortex, especially the orbitofrontal cortex, are related to BPSD symptoms, particularly apathy and psychosis [48]. Furthermore, gray matter volume loss in the medial orbital frontal cortex was seen in patients with frontotemporal lobar degeneration, with symptoms of disinhibition [49]. As part of the temporo-amygdala-orbitofrontal network, the orbitofrontal cortex may be associated with semantic deficits, language difficulties, personality changes, aggression, and disinhibition in patients with advanced AD [50]. Effects on other brain regions are also likely to be related to the potential effects of acupuncture on BPSD. The pontine tegmentum contains numerous serotonergic neurons, and abnormalities in this region could potentially be related to low mood and low self-esteem in patients with AD [51]. White matter changes in the frontal or parieto-occipital region and basal ganglia may be associated with psychotic symptoms, particularly delusional misidentification, in AD patients [52]. Modulation of some neurotransmitters may also explain the potential therapeutic effect of acupuncture on BPSD, and a recent review summarized the effect of acupuncture on glutamatergic neurotransmission in depression, anxiety, schizophrenia, and AD [53]. According to the results of the review [53], glutamatergic neurotransmission may be a common therapeutic pathway of acupuncture for BPSD symptoms (i.e., depression, anxiety, and psychotic symptoms) as well as the pathology (i.e., glutamate excitotoxicity) of AD. However, further research in this field is still needed, and the therapeutic mechanism of acupuncture for BPSD in AD patients has not been fully elucidated.

Although this systematic review allowed all of the original clinical studies that reported the effect of acupuncture on BPSD regardless of the study type, the fact that only seven studies were included suggests that research in this field is lacking. Therefore, in terms of EBM, to formally recommend that acupuncture be used as a supplement to BPSD management, more high-quality research in this field is needed in the future. In particular, dementia is currently becoming a social, national, and global problem beyond the individual or family level, so research support for non-pharmacological treatments including acupuncture for BPSD management of dementia patients is needed at the national level.

### 4.3. Limitations

This review presents some promising results of acupuncture as an adjuvant therapy for BPSD, but the following limitations should be considered:(1)Because the number of studies included in this review was small, quantitative synthesis was limited, and the results for each outcome were dependent on either one or two RCTs. In this situation, each RCT was implemented on a small scale, which may cause small-study effects, and its methodological quality is poor. Therefore, the reliability of the results obtained in this review was limited, and the level of evidence could be evaluated as weak.(2)The subgroup analysis planned in this review protocol [21] was not performed because of the lack of included studies. However, the characteristics of the participants and interventions in the included studies were not sufficiently homogeneous, so differences in effect estimates according to the severity of dementia, the severity of BPSD, and the duration of treatment should be further investigated in future studies.(3)The included studies also lacked the homogeneity of the evaluation tools used for BPSD evaluation. In particular, some studies used evaluation tools for individual BPSD symptoms such as HAMD, STAI, and PSQI and did not use an evaluation tool specific to BPSD such as BEHAVE-AD or NPI. Therefore, future studies require the uniformity of these evaluation tools, and the use of evaluation tools specific to BPSD is recommended.(4)Most of the included studies were conducted in China. In particular, all the included RCTs were implemented in China. Although publication bias using funnel plots was not evaluated in this review, studies implemented only in certain countries could potentially contribute to publication bias. In addition, as China has been using acupuncture for a long time, participants of acupuncture studies usually exhibit a favorable attitude toward this treatment method. These factors can act as obstacles to generalizing the results of this review to other countries.

## 5. Conclusions

Our review highlights the limited evidence supporting the effectiveness and safety of acupuncture for BPSD in patients with AD. Although some clinical studies have reported the potential benefits of acupuncture as an adjuvant therapy in managing BPSD or improving ADL, the evidence is weak and based on small studies. Although the development of effective non-pharmaceutical therapies for the management of BPSD is important, high-quality research on acupuncture seems to be lacking, and research in this field is needed.

## Figures and Tables

**Figure 1 jcm-10-03087-f001:**
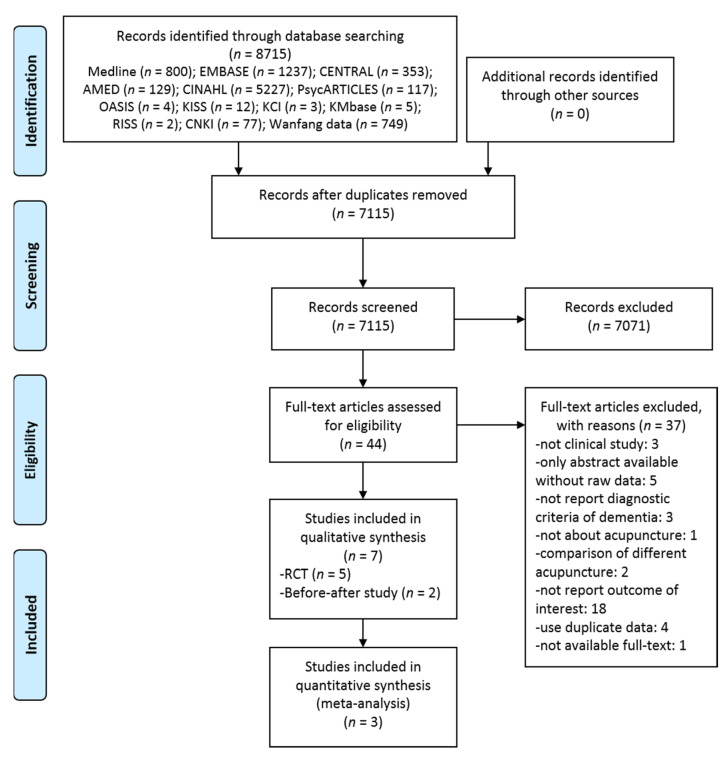
A PRISMA flow diagram of the literature screening and selection processes. AMED, Allied and Complementary Medicine Database; CENTRAL, Cochrane Central Register of Controlled Trials; CINAHL, Cumulative Index to Nursing and Allied Health Literature; CNKI, China National Knowledge Infrastructure; KCI, Korea Citation Index; KISS, Korean Studies Information Service System; KMbase, Korean Medical Database; OASIS, Oriental Medicine Advanced Searching Integrated System; RCT, randomized controlled trial; RISS, Research Information Service System.

**Figure 2 jcm-10-03087-f002:**
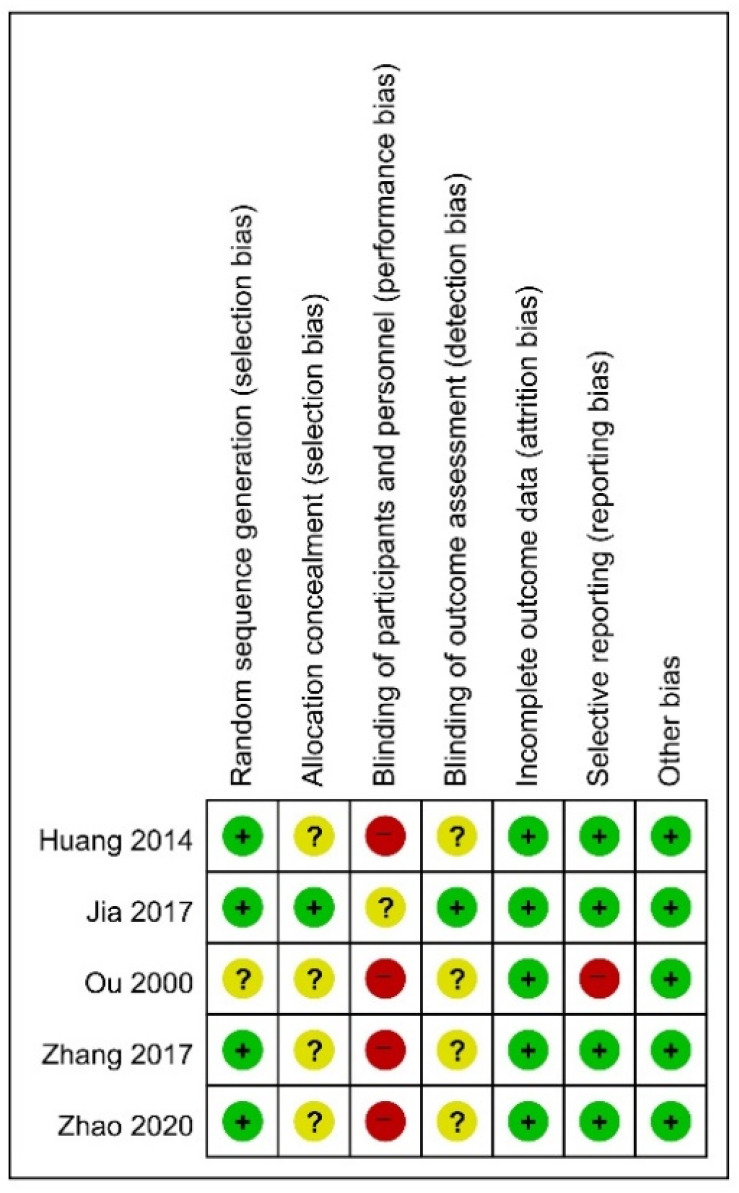
Risk of bias for all included studies. Low, unclear, and high risk, respectively, are represented with the following symbols: “+,” “?”, and “−”.

**Figure 3 jcm-10-03087-f003:**
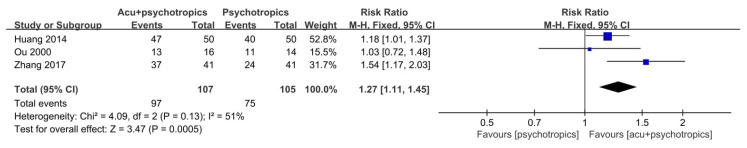
Forest plot of total effective rate comparing acupuncture plus psychotropic drugs and psychotropic drugs alone.

**Table 1 jcm-10-03087-t001:** Characteristics of included studies.

Study, Year, [Reference]	Sample Size(Included→Analyzed)	Mean Age (yr)	Sex (M:F)	Population	Intervention	Treatment Duration/F/U	Outcome
Ou, 2000, [35]	30(16:14) →30(16:14)	TG: 65.5 ± 6.8 CG: 64.7 ± 7.6	TG: 16(10:6) CG: 14(9:4)	-AD (mild, moderate) -baseline BPRS TG: 42.85 ± 5.25 CG: 41.91 ± 4.88	TG: EA + perphenazine (4–30 mg/d) CG: perphenazine (8–40 mg/d)	8 wk/none	1. TER (BPRS) 2. BPRS 3. CGI
Huang 2014, [36]	100(50:50) →100(50:50)	TG: 70.9 ± 8.6 CG: 71.5 ± 7.9	TG: 50(24:26) CG: 50(23:27)	-AD (mild, moderate, severe) -baseline HAMD TG: 22.30 ± 6.93 CG: 21.45 ± 7.01	TG: SA + fluoxetine (20 mg/d) CG: fluoxetine (20 mg/d)	8 wk/none	1. TER (HAMD) 2. HAMD 3. ADL
Jia 2017, [37]	87(43:44) →80(41:39)	TG: 75.11 ± 6.53 CG: 74.50 ± 6.83	TG: 43(13:30) CG: 44(16:28)	-AD (mild, moderate) -baseline NPITG: 9.28 ± 2.49 CG: 8.97 ± 2.69	TG: MA CG: donepezil (first 4 weeks: 5 mg/d; thereafter: 10 mg/d)	12 wk/12 wk	1. ADAS-cog 2. CIBIC + 3. ADCS-ADL23 4. NPI
Zhang 2017, [38]	82(41:41) →82(41:41)	TG: 66.12 ± 11.33 CG: 65.25 ± 10.62	TG: 41(23:18) CG: 41(22:19)	-AD -baseline PSQI TG: 14.12 ± 2.12 CG: 14.91 ± 3.32	TG: EA + midazolam (7.5 mg/d) CG: midazolam (7.5 mg/d)	30 d/none	1. TER (PSQI) 2. PSQI
Zhao 2020, [39]	120(60:60) →120(60:60)	TG: 72.0 ± 10.9 CG: 70.9 ± 11.2	TG: 60(30:30) CG: 60(28:32)	-AD -baseline MMSE TG: 14.34 ± 2.87 CG: 14.62 ± 3.01	TG: EA + passive music therapy + routine care (routine nursing, psychological comfort, daily cognitive training, diet modification) CG: routine care	4 wk/none	1. TER (MMSE, MoCA, SF-36) 2. MMSE 3. MoCA 4. ADL 5. BEHAVE-AD 6. SF-36 7. Levels of acetylcholine, choline acetylase, acetylcholinesterase, norepinephrine, 5-HT, and dopamine
Lombardo 2001, [40]	11→11	76	11(3:8)	-AD or VD -baseline MMSE 21.9 ± 5.2	MA	9–12 wk/none	1. POMS-anxiety 2. STAI-state anxiety 3. CSDD 4. GDS 5. MMSE 6. Boston naming test 7. Controlled oral word association test 8. Caregiver: SF-36-vitality, anxiety, depression 9. Patient: SF-36-vitality, anxiety, depression
Ying 2006, [41]	33→30	74.96 ± 6.41	30(12:18)	-AD -baseline MMSE 12.47 ± 2.62 -baseline BEHAVE-AD 18.90 ± 6.67	MA	8 wk/none	1. BEHAVE-AD 2. MMSE 3. ADL 4. TCM symptom score

Abbreviations. AD, Alzheimer’s disease; ADCS-ADL23, Alzheimer’s disease cooperative study—activities of daily living, 23-item scale; ADL, activities of daily living; BEHAVE-AD, the behavior pathology in Alzheimer’s disease rating scale; BPRS, the brief psychiatric rating scale; CG, control group; CGI, clinical global impression; CSDD, Cornell scale for depression in dementia; EA, electro-acupuncture; GDS, geriatic depression scale; HAMD, the Hamilton depression rating scale; MA, manual acupuncture; MMSE, mini-mental state examination; MoCA, Montreal cognitive assessment; NPI, the Neuropsychiatric inventory; POMS, profile of mood states; PSQI, the Pittsburgh sleep quality index; SF-36, short form 36 health survey; STAI, state-trait anxiety inventory; TCM, traditional Chinese medicine; TER, total effective rate; TG, treatment group; VD, vascular dementia. Note. Among outcomes, those that have been bolded fall within the scope of this systematic review.

**Table 2 jcm-10-03087-t002:** Methods of acupuncture performed.

Study, Year, [Reference]	Type of Acupuncture	Acupoints	Stimulation Method	Needle Retention Time	Treatment Frequency
Ou, 2000, [35]	EA	GV20, Yintang, GV14	De qi GV20 to Yintang or GV20 to GV14 Wave: continuous wave; Frequency: 2–4 Hz; Intensity: visible twitching of the local muscles, but comfortable and tolerable.	30 min	6 session/wk
Huang 2014, [36]	SA	anterior vertex zone, frontal zone	Manual stimulation	NR	6 session/wk
Jia 2017, [37]	MA	-Basic acupoints: CV17, CV12, CV6, ST36, TW5, SP10-Additional acupoints: LV3, GB39, ST40, BL17, ST44, ST25, CV4	De qi	30 min	3 session/wk
Zhang 2017, [38]	EA	GV20, GV24, EX-HN1, Anmian, Taiyang, P6, HT7, SP6, KI1	De qi Taiyang to EX-HN1 Wave: dilatational wave; Frequency: 2–100 Hz; Intensity: 2–4 V	25 min	1 session/d for 10d and rest for 3d
Zhao 2020, [39]	EA	GV20, BL23	Not reporting on De qi Around GV20 to GV20, around BL23 to BL23 Wave: continuous wave; Frequency: 50 Hz; Intensity: 2 V, 1 mA	20 min	1 session/d for 7d and rest for 1d
Lombardo 2001, [40]	MA	-Basic acupoints: GB9, GV16, GV20, GV23, GV24, PC6, HT7, SP6, EX-HN1, Yintang -Additional acupoints: ST36, LI4, GB20, GV17, SP4, KI3, SI3, BL62, BL23, GV26, EX-B2	NR	30 min	3 session/wk (1–2 wk), 2–3 session/wk (additional 7–10 wk)
Ying 2006, [41]	MA	-Basic acupoints: GV24, HT7, GV20, GV16, GV14, EX-B2 -Additional acupoints: BL23, GB39, LV3, SP6, HT5, HT7, ST36, SP10, BL17	De qi Manual stimulation every 10 min during needle retention	30 min	1 session/d for 6d and rest for 1d

Abbreviations. EA, electro-acupuncture; MA, manual acupuncture; NR, not recorded; SA, scalp acupuncture.

## Data Availability

The data extracted from included studies and data used for all analyses were all included in this manuscript.

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
