# Peer review of "Acupuncture for Behavioral and Psychological Symptoms of Dementia: A Systematic Review and Meta-Analysis"

_jcm, 2021, doi:10.3390/jcm10143087_

Round 1

Reviewer 1 Report

The review is well organized and structured although the number of studies included is small and limited to one country mainly. Following are some revisions:

  • The manuscript includes a meta-analysis so editing the title is suggested to indicate this.
  • The authors discussed dementia in all sections including the title, introduction, discussion and conclusion but all the studies that were included in the analysis has the same etiology which is AD. Since the underlying cause is an important factor that might affect the response to the treatment, I would suggest focusing the language of the manuscript on AD; specifically,  when discussing and making conclusions and in the title.
  • Edit some punctuations. Examples:  
    • Line 49: Delete the comma before “although”
    • Line 53: Delete the quotation mark in “ 'overall”
    • Line 23: add comma after “managing BPSD” and before “the evidence”

Author Response

  • Response to Comments from Reviewer 1

Comment 1:

The review is well organized and structured although the number of studies included is small and limited to one country mainly.

Response 1:           

Thank you for your careful review and insightful comments that have significantly enhanced our manuscript.

Comment 2:

Following are some revisions:

The manuscript includes a meta-analysis so editing the title is suggested to indicate this.

Response 2:           

Thank you for your comment. We have added a meta-analysis to the title of the manuscript in response to this comment.

“Acupuncture for behavioral and psychological symptoms of dementia: a systematic review and meta-analysis”

(Please refer to the red text on page 1.)

Comment 3:

The authors discussed dementia in all sections including the title, introduction, discussion and conclusion but all the studies that were included in the analysis has the same etiology which is AD. Since the underlying cause is an important factor that might affect the response to the treatment, I would suggest focusing the language of the manuscript on AD; specifically, when discussing and making conclusions and in the title.

Response 3:           

Thank you for your comment. In this revised manuscript, we further emphasized the expression ‘Alzheimer's disease’ in the Discussion and Conclusion sections, in response to this comment. In particular, in the Discussion section, we added explanations based on published studies on the relationship between acupuncture and BPSD in patients with AD.

However, in the case of title, it was described that this systematic review was intended to investigate BPSD of dementia, not limited to AD, in our pre-registered protocol as well as published protocol paper. Therefore, in the revised manuscript, ‘AD’ was difficult to add to the title in addition to the behavioral and psychological symptoms of dementia, which is the full term for BPSD.

“This systematic review is the most comprehensive review and meta-analysis to date conducted to analyze the effectiveness, safety, and research status of acupuncture for BPSD. A total of seven clinical studies, including five RCTs [35-39] and two before-after studies [40,41], mainly on patients with AD, were included in this review.”

(Please refer to the red text on page 14.

“Although the underlying therapeutic mechanism of acupuncture for BPSD in patients with AD has not been fully elucidated, the results of pre-clinical and clinical experiments can be referred to. Lu (2017) found that manual acupuncture on ST36 for 30 days in a rat AD model activated brain areas including the orbital cortex, medulla oblongata, and pontine tegmentum [45]. Moreover, Shan (2018) found that manual acupuncture on LI4 and LR3 in patients with AD was related to enhanced activations in cognitive-related areas, including the inferior frontal gyrus, as well as sensorimotor-related areas, the basal ganglia, and the cerebellum, by using functional magnetic resonance imaging (fMRI) [46]. A review investigating changes in acupuncture-related brain activity in fMRI concluded that acupuncture stimuli could promote activity changes in a wide range of brain regions, including the somatosensory cortices, limbic system, basal ganglia, brain stem, and cerebellum [47]. Acupuncture-induced changes in local brain activity in AD patients may be related to the therapeutic effect of acupuncture on BPSD. Among them, the possible impact of acupuncture on the orbitofrontal cortex seems to be promising. A previous systematic review suggested that volume reductions or decreased metabolism in the orbital cortex, especially the orbitofrontal cortex, are related to BPSD symptoms, particularly apathy and psychosis [48]. Furthermore, gray matter volume loss in the medial orbital frontal cortex was seen in patients with frontotemporal lobar degeneration, with symptoms of disinhibition [49]. As part of the temporo-amygdala-orbitofrontal network, the orbitofrontal cortex may be associated with semantic deficits, language difficulties, personality changes, aggression, and disinhibition in patients with advanced AD [50]. Effects on other brain regions are also likely to be related to the potential effects of acupuncture on BPSD. The pontine tegmentum contains numerous serotonergic neurons, and abnormalities in this region could potentially be related to low mood and low self-esteem in patients with AD [51]. White matter changes in the frontal or parieto-occipital region and basal ganglia may be associated with psychotic symptoms, particularly delusional misidentification, in AD patients [52]. Modulation of some neurotransmitters may also explain the potential therapeutic effect of acupuncture on BPSD, and a recent review summarized the effect of acupuncture on glutamatergic neurotransmission in depression, anxiety, schizophrenia, and AD [53]. According to the results of the review [53], glutamatergic neurotransmission may be a common therapeutic pathway of acupuncture for BPSD symptoms (i.e., depression, anxiety, and psychotic symptoms) as well as the pathology (i.e., glutamate excitotoxicity) of AD. However, further research in this field is still needed, and the therapeutic mechanism of acupuncture for BPSD in AD patients has not been fully elucidated.”

(Please refer to the red text on pages 15-16.)

“Our review highlights the limited evidence supporting the effectiveness and safety of acupuncture for BPSD in patients with AD. Although some clinical studies have reported the potential benefits of acupuncture as an adjuvant therapy in managing BPSD or improving ADL, the evidence is weak and based on small studies. Although the development of effective non-pharmaceutical therapies for the management of BPSD is important, high-quality research on acupuncture seems to be lacking, and research in this field is needed.”

(Please refer to the red text on page 17.)

Comment 4:

Edit some punctuations. Examples:

Line 49: Delete the comma before “although”

Line 53: Delete the quotation mark in “ 'overall”

Line 23: add comma after “managing BPSD” and before “the evidence”

Response 4:           

Thank you for your careful comment. The punctuations were all edited, in response to this comment.

Reviewer 2 Report

The present work is  on the effects of acupuncture on symptoms in demented patients. A review and  metaanalysis of studies published previously is performed. Benefital effects of acupuncture are noted when used as a blended care approach. The topic is of interest in view of the public health impact of dementia. The methods used are sound. The results are presented in an adequate way. However the manuscript may be improved considering the following two points.

1st It should be mentioned which specific symptoms  of demented patients are improved by ad on acupuncture i.e. is working memory positively influence ?

2nd The introduction and discussion sections  widely deal with statistical and technical aspects of the  study data analyzed. This deems inadequate in view of the limited knowledge on  acupuncture as a treatment regimen in various diseases.   Both sections should deal with the potential effects of acupuncture in more depth and detail.  Especially the mechanism of action of acupuncture should be mentioned. It should be explained why acupuncture is hypothesized to improve behavioral or psychological symptoms in patients suffering from dementia.    

Author Response

  • Response to Comments from Reviewer 2

Comment 1:

The present work is on the effects of acupuncture on symptoms in demented patients. A review and metaanalysis of studies published previously is performed. Benefital effects of acupuncture are noted when used as a blended care approach. The topic is of interest in view of the public health impact of dementia. The methods used are sound. The results are presented in an adequate way.

Response 1:           

Thank you for your careful review and insightful comments that have significantly enhanced our manuscript.

Comment 2:

However the manuscript may be improved considering the following two points.

1st It should be mentioned which specific symptoms of demented patients are improved by ad on acupuncture i.e. is working memory positively influence ?

Response 2:           

Thank you for your comment. We added descriptions of individual BPSD symptoms reported to have improved after acupuncture in the Discussion section.

“In summary, in the included studies, improvements in individual BPSD after acupuncture were reported, in particular, for depression, anxiety, sleep quality, paranoid and delusional ideation, and activity disturbances. Among them, the improved individual symptoms of BPSD after acupuncture in the included RCTs were depression and sleep quality.”

(Please refer to the red text on page 15.)

Comment 3:

2nd The introduction and discussion sections widely deal with statistical and technical aspects of the study data analyzed. This deems inadequate in view of the limited knowledge on acupuncture as a treatment regimen in various diseases. Both sections should deal with the potential effects of acupuncture in more depth and detail. Especially the mechanism of action of acupuncture should be mentioned. It should be explained why acupuncture is hypothesized to improve behavioral or psychological symptoms in patients suffering from dementia.

Response 3:           

Thank you for your comment. We strongly agree that our description of potential effects of acupuncture in the original manuscript was not deep and detailed discussion. Therefore, in this revised manuscript, both the Introduction and Discussion sections have strengthened the explanation of the possible acupuncture mechanism for BPSD.

“Acupuncture, a promising non-pharmacological intervention, is recommended worldwide for some conditions such as pain [12] although it is not evaluated to have robust evidence to be officially recommended for the management of BPSD [13]. Acupuncture has been studied not only for pain conditions, but also for a wide range of health conditions such as neoplasms/cancer, pregnancy or labor, mood disorders, stroke, nausea/vomiting, sleep, and paralysis/palsy [14], and has the potential to improve overall health [15]. In addition, this modality appears to be generally safe [16]. In this context, acupuncture can be considered a promising treatment for the elderly population. Psychiatric manifestations in patients with dementia have been well documented in neuroimaging and neuropathology studies [17]. The effects of acupuncture on the nervous system of human subjects and animal models with neurological diseases including AD have also been reported [18]. It has been stated that acupuncture can influence AD pathology through the inhibition of accumulation of toxic proteins, regulation of glucose metabolism, reduction of neuronal apoptosis, and neuroprotective effects [18]. Therefore, this treatment may have a therapeutic effect on BPSD, which accompanies AD pathology [19,20].”

(Please refer to the red text on page 2.)

“Although the underlying therapeutic mechanism of acupuncture for BPSD in patients with AD has not been fully elucidated, the results of pre-clinical and clinical experiments can be referred to. Lu (2017) found that manual acupuncture on ST36 for 30 days in a rat AD model activated brain areas including the orbital cortex, medulla oblongata, and pontine tegmentum [45]. Moreover, Shan (2018) found that manual acupuncture on LI4 and LR3 in patients with AD was related to enhanced activations in cognitive-related areas, including the inferior frontal gyrus, as well as sensorimotor-related areas, the basal ganglia, and the cerebellum, by using functional magnetic resonance imaging (fMRI) [46]. A review investigating changes in acupuncture-related brain activity in fMRI concluded that acupuncture stimuli could promote activity changes in a wide range of brain regions, including the somatosensory cortices, limbic system, basal ganglia, brain stem, and cerebellum [47]. Acupuncture-induced changes in local brain activity in AD patients may be related to the therapeutic effect of acupuncture on BPSD. Among them, the possible impact of acupuncture on the orbitofrontal cortex seems to be promising. A previous systematic review suggested that volume reductions or decreased metabolism in the orbital cortex, especially the orbitofrontal cortex, are related to BPSD symptoms, particularly apathy and psychosis [48]. Furthermore, gray matter volume loss in the medial orbital frontal cortex was seen in patients with frontotemporal lobar degeneration, with symptoms of disinhibition [49]. As part of the temporo-amygdala-orbitofrontal network, the orbitofrontal cortex may be associated with semantic deficits, language difficulties, personality changes, aggression, and disinhibition in patients with advanced AD [50]. Effects on other brain regions are also likely to be related to the potential effects of acupuncture on BPSD. The pontine tegmentum contains numerous serotonergic neurons, and abnormalities in this region could potentially be related to low mood and low self-esteem in patients with AD [51]. White matter changes in the frontal or parieto-occipital region and basal ganglia may be associated with psychotic symptoms, particularly delusional misidentification, in AD patients [52]. Modulation of some neurotransmitters may also explain the potential therapeutic effect of acupuncture on BPSD, and a recent review summarized the effect of acupuncture on glutamatergic neurotransmission in depression, anxiety, schizophrenia, and AD [53]. According to the results of the review [53], glutamatergic neurotransmission may be a common therapeutic pathway of acupuncture for BPSD symptoms (i.e., depression, anxiety, and psychotic symptoms) as well as the pathology (i.e., glutamate excitotoxicity) of AD. However, further research in this field is still needed, and the therapeutic mechanism of acupuncture for BPSD in AD patients has not been fully elucidated.”

(Please refer to the red text on pages 15-16.)

Round 2

Reviewer 2 Report

well done